# A Population Pharmacokinetic and Pharmacodynamic Model of CKD-519

**DOI:** 10.3390/pharmaceutics12060573

**Published:** 2020-06-19

**Authors:** Choon Ok Kim, Sangil Jeon, Seunghoon Han, Min Soo Park, Dong-Seok Yim

**Affiliations:** 1Department of Clinical Pharmacology, Severance Hospital, Yonsei University College of Medicine, Seoul 03722, Korea; delivery98@yuhs.ac (C.O.K.); minspark@yuhs.ac (M.S.P.); 2Qfitter Inc., Seoul 06578, Korea; si.jeon@qfitter.com (S.J.); waystolove@catholic.ac.kr (S.H.); 3Department of Pharmacology, Seoul St. Mary’s Hospital, College of Medicine, The Catholic University of Korea, Seoul 06591, Korea; 4PIPET (Pharmacometrics Institute for Practical Education and Training), College of Medicine, The Catholic University of Korea, Seoul 06591, Korea; 5Department of Pediatrics, Yonsei University College of Medicine, Seoul 03722, Korea

**Keywords:** population PK-PD modeling, CKD-519, CETP inhibitor, CETP activity, HDL-C

## Abstract

CKD-519 is a selective and potent cholesteryl ester transfer protein (CETP) inhibitor that is being developed for dyslipidemia. Even though CKD-519 has shown potent CETP inhibition, the exposure of CKD-519 was highly varied, depending on food and dose. For highly variable exposure drugs, it is crucial to use modeling and simulation to plan proper dose selection. This study aimed to develop population pharmacokinetic (PK) and pharmacodynamics (PD) models of CKD-519 and to predict the proper dose of CKD-519 to achieve target levels for HDL-C and LDL-C using results from multiple dosing study of CKD-519 with a standard meal for two weeks in healthy subjects. The results showed that a 3-compartment with Erlang’s distribution, followed by the first-order absorption, adequately described CKD-519 PK, and the bioavailability, which decreased by dose and time was incorporated into the model (NONMEM version 7.3). After the PK model development, the CETP activity and cholesterol (HDL-C and LDL-C) levels were sequentially modeled using the turnover model, including the placebo effect. According to PK-PD simulation results, 200 to 400 mg of CKD-519 showing a 40% change in HDL-C and LDL-C from baselines was recommended for proof of concept studies in patients with dyslipidemia.

## 1. Introduction

Cholesteryl ester transfer protein (CETP) is a hydrophobic plasma glycoprotein consisting of 476 amino acids with a mass of 53 kDa [1]. It mediates bidirectional transfers of cholesteryl ester and triglyceride between high-density lipoprotein cholesterol (HDL-C) and apolipoprotein B-containing lipoproteins, such as very-low-density lipoprotein and low-density lipoprotein cholesterol (LDL-C) [2]. It has been reported that CETP plays a key role in atherosclerosis through its impact on lipid metabolism [1]. Thus, there have been many efforts to modify the composition of the plasma lipoprotein favorably through inhibition of CETP activity [3].

CKD-519 is a potent, selective CETP inhibitor being developed for the treatment of dyslipidemia. It is one of the cycloalkenyl benzene derivatives with a molecular weight of 601.60 g/mol (C_31_H_34_F_7_NO_3_) [4]. It is poorly soluble, with aqueous solubility of less than 1.0 μg/mL [4]. In the in vitro solubility test, CKD-519 was insoluble in deionized water and freely soluble in ethyl alcohol with its LogD of 7.4, and LogP is >5.0 at pH 7.4 [4,5].

CKD-519 exhibited a maximum of 63–83% inhibitory effect of CETP activity after single administration with 25–400 mg in healthy subjects (EC_50_ 17.3 ng/mL) [5]. The exposure of CKD-519 increased with the dose. However, maximum plasma concentration (C_max_) and area under the plasma concentration-time curve (AUC) normalized by dose decreased with each incremental dose [5]. In addition, the pharmacokinetics (PK) of CKD-519 exhibited significant variation in the fasted and fed states. Administration of single CKD-519 200 mg with a standard meal resulted in an 8.7 and 5.8 fold increase in C_max_ and AUC relative to the fasted state, respectively (in house data). The inhibitory effect of CETP activity by CKD-519 200 mg was also higher in the fed (91.4%) than in the fasted group (81.5%).

Despite potent CETP inhibition, the exposure of CKD-519 varied by food and dose. This study aimed to characterize the population PK and pharmacodynamics (PD) of CKD-519 using multiple dosing study of CKD-519 with a standard meal in healthy subjects. The PK-PD model can be applied to the design of proof of concept studies to observe HDL-C and LDL-C changes in patients.

## 2. Materials and Methods

### 2.1. Ethics

The protocols were approved by the institutional review board of Severance Hospital, Yonsei University College of Medicine (Seoul, Korea; IRB number 4-2015-0126 and 4-2016-0949, 02 April 2015 and 17 December 2016). They were also registered at the clinicaltrals.gov (identifier: NCT02753504 and NCT03210649, 28 April 2016 and 07 July 2017). These studies were carried out in accordance with the relevant regulatory requirements, including the Declaration of Helsinki and the Korean good clinical practice. All subjects provided written informed consent before enrollment in this study.

### 2.2. Study Design

A randomized, double-blinded, placebo-controlled, multiple-dose, and dose-escalation study was conducted in a total of 32 healthy male subjects aged 19–55 years in four cohorts (50, 100, 200, and 400 mg). Six subjects were assigned to CKD-519, and two to matching placebo in each cohort. Dose escalation to the next cohort was done after the safety profile was reviewed. CKD-519 was available as two dosage forms of 50 and 100 mg tablets.

Enrolled subjects were admitted on day −1. CKD-519 or placebo was administered to randomly assigned subjects for 14 days with a standard breakfast meal (700–800 kcal, containing 5–25% fat content). All subjects were hospitalized until completion of blood sampling for PK and PD (from day −1 to day 21). All meals were nutritionally equivalent and provided at the same time scheduled throughout the study period. The participants were not permitted to take any drugs or herbal medications and also were restricted from consuming any other snacks or beverages during the study.

Blood samples for PK or PD were collected on days 1 (0, 1, 2, 3, 4, 5, 6, 7, 8, 10, 12, 18, and 24 h) and 14 (0, 1, 2, 3, 4, 5, 6, 7, 8, 10, 12, 18, 24, 32, 48, 72, 96, 120, 144, and 168 h), along with daily sampling on days 2 through 13 to determine trough levels of drug concentration, CETP activity, and lipid values. Each blood sample for PK and CETP activity was collected in sodium heparin tubes and then centrifuged at 1900 *g* for 10 min. Their aliquots were stored at or below 70 °C until analysis. The plasma concentrations of CKD-519 were measured using a validated high-performance liquid chromatography-tandem mass spectrometry assay [5]. A 100 μL plasma sample was mixed with 10 μL internal standard working solution and 300 μL acetonitrile. After centrifugation, 3 μL supernatant was injected into the column. The lower limit of quantification was 1.0 ng/mL. The calibration curve was linear over the concentration range of 1.0–2000 ng/mL. The precision of the assay was less than 2.7% coefficient of variation, and the accuracy of the assay was within the range of 93.7–113.0% [5]. Plasma CETP activity was measured by using a fluorescent assay method previously described [5]. HDL-C and LDL-C were measured with an ADIVA 1650 Clinical Chemistry System (Siemens Medical Solutions, Tarrytown, NY, USA).

### 2.3. Model Development

PK and PD models were developed using NONMEM version 7.3 (Icon Development Solutions, Ellicott City, MD, USA) using a first-order conditional estimation with interaction method (FOCE-INTER). R (version 3.6.1, R Foundation for Statistical Computing, Vienna, Austria) was used for data preparation, graphical analysis, and model diagnostics. The population PK-PD model was developed sequentially. The PK model was initially developed, and then the CETP model (PK-PD) was developed using individual PK parameters estimated from the final PK model. Finally, the cholesterol (HDL-C or LDL-C) models were developed using the individual CETP parameters estimated from the final CETP model.

Inter-individual variability (IIV) for each parameter in all models was assumed to follow a Gaussian distribution with a mean of 0 and variance of ω^2^. For the PK model, once the IIV term was selected, the covariance between the terms was assessed using an omega block on selected parameters. As for the residual error, additive, proportional, and combined models were tested. The model was selected after a comparison of several criteria such as a decrease in the objective function value (OFV), completion of the estimation and covariance routines, the precision of the parameter and error estimates, and visual exploration of the diagnostic plots.

Once the structural model was evaluated, covariate analysis was performed using the stepwise forward selection and backward elimination process. Statistical significance was defined as a decrease in the OFV by more than 3.84 for forward selection and an increase in the OFV of at least 6.635 (chi-square distribution, 1 degree of freedom). The following factors were evaluated as covariates: age, weight, body mass index, creatinine clearance calculated from the modification of diet in renal disease equation, and alanine aminotransferase. They were considered significantly associated with the parameters if both of the OFV decrease and physiological plausibility were satisfied.

#### 2.3.1. Population PK Model

Several disposition models were assessed, including 1-, 2-, and 3- compartment structural models. Absorption was evaluated with first-order, zero-order (with and without lag time), Weibull-type, and Erlang’s model [6]. The bioavailability was assessed about the effects of the dose and time. Drug elimination was assumed to follow first-order kinetics.

#### 2.3.2. Population CETP Model

Since CETP inhibition is closely related to HDL-C increase and LDL-C decrease, the CETP model was included as a link between the PK model of CKD-519 and cholesterol (HDL-C and LDL-C) models [7]. For the CETP model, CETP activity was used as a dependent variable as this was the primary parameter of interest.

The turnover model was used to describe the relationship between CETP activity and plasma concentration of CKD-519. The effect of CKD-519 was applied in the model as stimulating the elimination of CETP activity. Because the CETP activity increased over time in the placebo group, the response of placebo treatment was incorporated into the CETP model. The effects of CKD-519 and placebo treatment were evaluated as linear, E_max_, or sigmoid E_max_ relationships.

#### 2.3.3. Population HDL-C and LDL-C Models

For each of the HDL-C and LDL-C model development, their clinical laboratory results were used. Their responses by changing CETP activity were described using turnover models. Taking into account the physiological mechanism of CETP on lipid metabolism, the CETP effect was incorporated into the model as a stimulatory function of HDL-C elimination and an inhibitory function of LDL-C elimination. The effect of CETP activity on HDL-C or LDL-C was evaluated using linear, E_max_, or sigmoid E_max_ models.

### 2.4. Model Evaluation

A bootstrap resampling method was used to evaluate the stability and reliability of the final models. Bootstrap of 1000 resamplings was performed for each final model of PK, CETP, and HDL-C/LDL-C model to obtain the median and 95% confidence intervals for all parameters. In addition, the visual predictive check (VPC) with a total of 1000 simulated datasets was conducted on each PK, CETP, and HDL-C/LDL-C model. Results from the VPC were graphically assessed by comparing it with the overlaid observed data.

### 2.5. Simulation

The simulation was performed using the final model to guide the proper dose selection for an increase of HDL-C and decrease of LDL-C to the target levels. The target HDL-C and LDL-C levels were set by 40% changes compared to the baselines [8]. To simulate mean HDL-C and LDL-C values after three weeks of daily CKD-519 dosing, parameters of population fixed effects were drawn from final models. IIV or residual variability were not included in these simulations.

## 3. Results

### 3.1. Demographics

A total of 32 male subjects were randomized and completed these trials. Demographic and baseline characteristics are summarized in Table 1.

Demographic and baseline characteristics were well balanced across treatment groups. The mean age was 32.2 years (19.0 to 47.0 years), and the mean body weight was 68.7 kg (58.3 to 79.5 kg). A total of 1392 points of plasma CKD-519 concentration and 2064 points of CETP activity samples were measured. As for the HDL-C and LDL-C, 656 samples were measured.

### 3.2. Population Pharmacokinetic Analysis

The disposition of CKD-519 was assumed to follow first-order characteristics. A 3-compartment model with Erlang’s distribution (with five sequential compartments) followed by the first-order absorption best described the concentration data of CKD-519. A schematic diagram of the final PK model is shown in Figure 1.

B_max_, the maximal effect of dose on bioavailability; BA_50_, the dose of half-maximal effect on bioavailability; α, the proportionality constant for the fraction of first-order absorption process depending on time; K_tr_, transit rate constant; K_a_, rate of constant for the first-order absorption; V/F, the volume of the central compartment; V2/F, the volume of peripheral compartment 1; V3/F, the volume of peripheral compartment 2; Q2/F, inter-compartmental clearance for compartment 1; Q3/F, inter-compartmental clearance for compartment 2; CL/F, clearance; Kin_base_, CETP activity production rate at baseline; K_max_, maximal effect of time of CETP activity production rate; K_50_, time of half-maximal effect on CETP production rate; K_out_, the first-order rate constant for the elimination of CETP activity; K_syn_, the production rate of HDL-C or LDL-C; K_deg_, elimination rate of HDL-C or LDL-C; β, the proportionality constant for the fraction of LDL-C production process; T, time.

The following parameters were estimated in the model: absorption rate constant (K_a_), transit rate constant (K_tr_), clearance (CL/F), central volume (V/F), peripheral compartment volumes (V2/F and V3/F), inter-compartmental clearances (Q2/F and Q3/F), and bioavailability (F).

A descriptive approach was used to describe the nonlinear exposure of CKD-519. The exposure of CKD-519 tended to decrease by increasing doses and repeated administration in the dose range of 50–200 mg. However, in the 400 mg group, its exposure after multiple administration resulted in a three-fold increase of AUC compared to that of the first dose. Therefore, the bioavailability terms were modeled using a combined term of dose and time as follows:(1)F=BA ×FT
(2)$$\mathrm{BA}=B_{\max}\times \left(1-\frac{\mathrm{DOSE}}{\left({\mathrm{BA}}_{50}+\mathrm{DOSE}\right)}\right)$$
(3)$$\mathrm{FT}={\;e}^{-\alpha \times \mathrm{TIME}}$$
where BA is the relative bioavailability depending on dose, FT is the time-dependent absorption fraction, B_max_ is the maximal effect of dose on bioavailability, BA_50_ is the dose of a half-maximal effect on bioavailability, and α is the proportionality constant for the fraction of absorption depending on time. The α for 50~200 mg (α_1_) was estimated in the model, but the α for 400 mg (α_2_) was fixed at zero.

Exponential IIV terms were included for CL/F, V3/F, B_max_, and K_tr_. Residual variability was represented by a proportional error. There was no statistically significant covariance included in the final model. Serum creatinine was identified as influential covariates for CL/F in the forward selection, but it did not satisfy backward elimination criteria. Covariate analysis was performed for the other variables, but none were significant. The final parameters estimated in the model are shown in Table 2.

Figure 2A shows the model diagnostic plots for the final PK model. Residual errors in the lower panels are evenly scattered over the range of predicted concentrations. The VPC obtained with a 1000 simulated datasets is presented in Figure 3A. The observed data were fairly well predicted by the final model.

Black line is line of identity, and gray line is locally weighted regression smooth line. IWRES, individual weighted residuals.

### 3.3. Population CETP Analysis

The change in the CETP activity was described using a turnover model:(4)$$\frac{{\mathrm{dCETP}}_{\mathrm{activity}}}{\mathrm{dt}}=K_{\mathrm{in},\mathrm{base}}\times \left(1+\mathrm{Placebo}\right)-K_{\mathrm{out}}\times \left(1+\mathrm{Drug}\right)$$
where K_in, base_ is the zero-order constant for the increase of CETP activity at baseline, placebo is the placebo response to CETP activity, K_out_ is the first-order rate constant for the decrease in CETP activity, and Drug is the effect of CKD-519 on CETP activity.

Several structural models for effects of drug and placebo were tested, including linear, E_max_, and sigmoid E_max_ models. Both of them were best described with E_max_ model. The placebo and drug effects were as follows, respectively:(5)$$\mathrm{Placebo}=\frac{K_{\max}\times \mathrm{time}}{K_{50}+\mathrm{time}}$$
(6)$$\mathrm{Drug}=\frac{E_{\max}\times C_{p}}{{\mathrm{EC}}_{50}+C_{p}}$$
where K_max_ is the maximal effect of time on CETP activity production rate, K_50_ is the time of half-maximal effect on CETP production rate, E_max_ is the maximal effect of plasma drug concentration on CETP activity, EC_50_ is the plasma concentration of half-maximal effect on CETP activity, and C_p_ is the plasma concentration of CKD-519.

The final model for CETP activity includes IIV on K_in,base_, and the maximum drug effect, and the combined error structure was used for residual variability. There were no significant covariates in the model. The final parameter estimates are presented in Table 3, and the goodness-of-fit plots of the final model is shown in Figure 2B. Overall, the results suggested a reasonable fit of the model to observed CETP activity.

### 3.4. Population HDL-C and LDL-C Analysis

HDL-C and LDL-C models were best described with the sigmoid E_max_ function. The LDL-C in the placebo group tended to increase gradually over time. Thus, a linear model was used to describe the placebo effect on LDL-C after comparison with E_max_ and sigmoid E_max_ models. The changes in HDL-C and LDL-C concentration were described as follows:(7)$$\frac{\mathrm{dHDL}}{\mathrm{dt}}=K_{\mathrm{syn}}-K_{\deg}\times A_{\mathrm{HDL}}\times \left(1+{\mathrm{RES}}_{\mathrm{CETP}}\right)$$
(8)$$\frac{\mathrm{dLDL}}{\mathrm{dt}}=K_{\mathrm{sy}}\times \left(1+\mathrm{PLA}\right)-K_{\deg}\times A_{\mathrm{LDL}}\times \left(1-{\mathrm{RES}}_{\mathrm{CETP}}\right)$$
(9)$${\mathrm{RES}}_{\mathrm{CETP}}=\frac{{\mathrm{CETP}}_{\mathrm{activity}}^{\gamma }\times R_{\max}}{{\mathrm{CETP}}_{\mathrm{activity}}^{\gamma }+R_{50}^{\gamma }}$$
(10)PLA=β×time
where K_syn_ and K_sy_ are the zero-order production rate constant, K_deg_ is the first-order elimination rate constant, RES_CETP_ is the change in CETP activity, R_max_ is the maximal effect of CETP activity on HDL-C and LDL-C response, R_50_ is the CETP activity required to attain 50% maximal response of HDL-C and LDL-C, PLA is the placebo effect, and β is the proportionality constant.

The final model of HDL-C included IIV on baseline HDL-C, and its residual error was additive. For the LDL-C model, IIV was used in the baseline LDL-C, β, and γ. The residual variability of LDL-C was in the combined structure. No covariates were found to be significant in the HDL-C and LDL-C models. The parameters estimated in the final model are shown in Table 4.

Figure 2C,D show the model diagnostic plots for our final HDL-C and LDL-C model, respectively. VPC from the 1000 simulated datasets are presented in Figure 3C,D. The VPC results showed that the 90% prediction intervals of the simulated data matched well with the observed data for both of the HDL-C and LDL-C.

### 3.5. Simulation

The final HDL-C and LDL-C model were used in population simulations to find out the proper dose to achieve target lipid levels. The simulations predicted HDL-C and LDL-C after 21 days of administration of CKD-519 with different doses (Figure 4).

As the dose increased, the HDL-C and LDL-C responses also increased. They showed maximum responses after repeated dosing for ten days. Because the final model explained observed data using a time-dependent decrease of bioavailability for 50–200 mg and time-dependent increase of CETP activity (placebo group), the simulated effects of CKD-519 on cholesterol decreased gradually afterward in 50–200 mg dose groups, but not in the 400 mg. The HDL-C increase of 40% and LDL-C decrease of 40% from baselines were observed in 200 mg and 400 mg groups.

## 4. Discussion

CKD-519 has shown variable systemic exposure depending on dose and food (in house data) [5]. It is necessary to understand the relationship between PK of CKD-519 and lipid response in order to find out an appropriate dose for the treatment of dyslipidemia. The PK-PD models of CKD-519 and HDL-C/LDL-C levels mediated by CETP activity were developed in this study in that context. At first, a PK/PD model was constructed to connect the CKD-519 exposure and CETP activity, and then, lipid responses (HDL-C and LDL-C levels) were further connected to the PK-CETP activity model.

In healthy subjects, CKD-519 exposure was found to increase in a less than dose-proportional manner with respect to dose over the 25–400 mg dose range that was studied [5]. In our report, less than dose-proportional increases in the CKD-519 exposure were also observed, with CL/F approximately 2.5-fold increase from 11.98 L/h at 50 mg to 30.04 L/h at 200 mg. However, CL/F for 400 mg was 23.98 L/h that was less than that of 200 mg. The observed terminal half-life did not appear to vary with dose, so the lack of dose-proportionality is thought to be the result of changes in the extent of absorption rather than changes in the rate of elimination as reported in other CETP inhibitors, anacetrapib, and evacetrapib [9]. The CKD-519 exposure decreased after multiple administration in the 50–200 mg dose range, but not in 400 mg. The saturation of the absorption pathway suggested in previous studies [5,10], or autoinduction [11] may be employed to account for the phenomena observed in 50–200 mg groups, but not in 400 mg. Thus, a descriptive model was applied to estimate relative bioavailability changes of CKD-519 with dose and time.

CKD-519 showed a tri-phasic disposition pattern and had a long half-life with 145.4–166.2 h. This feature was also similar in other CETP inhibitors [12]. Among them, anacetrapib was well known for its long terminal half-life. [13] Some studies reported that anacetrapib accumulates in the adipose tissue, and its concentration in blood declines very slowly [13,14]. Based on these findings, anacetrapib was also analyzed using a 3-compartment model [13]. Similar to anacetrapib, CKD-519 has neutral lipophilic properties, which explains its high affinity and accumulation in the adipose tissue [5]. Therefore, the PK model of CKD-519 employed a 3-compartment structure like in anacetrapib, which depicted the disposition of observed data better than 1- or 2-compartment structures.

Also, all dose groups of CKD-519 in our study have shown delayed absorption (T_max_ approximately 1 h). Hence, a few delayed absorption models were tested under the 3-compartment structure. The Weibull-type absorption model used in the animal model of CKD-519 was unstable and terminated without minimization [10]. When Erlang’s absorption model was finally applied, OFV was the lowest, and goodness-of-fit plots were the best compared to other delayed absorption models. Five transit compartments were added, and the OFV was the lowest at 8789.41, which was reduced by about 386 compared to the OFV of zero-order with lag time model. According to these results, a 3-compartment PK model of CKD-519 with Erlang’s absorption (with five sequential compartments) was finally chosen.

CETP activity was rapidly suppressed and sustained during repeated CKD-519 administration. On the other hand, CETP activity gradually increased over time in the placebo group. As reported earlier in other CETP inhibitors [15], the administration of CKD-519 in this study led to an approximately 2.5 fold increase in serum CETP concentration in a time-dependent manner (in house data). In the placebo group, serum CETP concentration increased by about 30% after two weeks (in house data). It has been reported that transcription of the CETP gene is under the control of extrinsic and intrinsic factors [16]. Dietary cholesterol up-regulates CETP expression in human CETP-transgenic mice [16], and plasma cholesterol levels also correlate with CETP mass in humans [17]. The increase in CETP activity in the placebo group might be due to increased serum CETP mass because dietary cholesterol regularly served during hospitalization for three weeks could have promoted CETP expression. Besides, total cholesterol, including LDL-C and triglyceride in the placebo group, also slowly increased, but their HDL-C levels were stable. Because increased plasma cholesterol could have up-regulated the CETP gene expression in the placebo group, the placebo effect term on CETP activity was tested in the PD modeling step. Linear, E_max_, and sigmoid E_max_ models were tested to explain CETP activity increase in the placebo group, and the E_max_ model by time best described the placebo effect.

HDL-C and LDL-C responses to CKD-519 were rather slow, whereas the CETP activity rapidly decreased and stayed at the nadir. Turnover models were used to describe the relationship between CETP activity and HDL-C/LDL-C responses. In addition, a placebo effect model aforementioned was also used for LDL-C. The maximal increase in HDL-C was estimated to be 160–190% and serum CETP activity exerting the half-maximal effect (R_50_) on HDL-C was 185.0 pmole. The maximal decrease in LDL-C was estimated to be 47–66%, and the corresponding R_50_ was 80.7 pmole.

Using the final PK-CETP-lipid model, we performed simulations to investigate the proper dose of CKD-519 to achieve target levels for HDL-C (40% increase) and LDL-C (40% decrease) as in the case of anacetrapib [8]. The simulation showed that 200 to 400 mg CKD-519 once daily will result in lipid-altering effects that are near the PD targets (dotted lines, Figure 4). In the case of 50–200 mg, the simulated responses decreased slowly after the tenth day of dosing because of the time dependency of bioavailability and CETP activity. As our PK and PD data were obtained from rather a small number of young, healthy subjects admitted for only three weeks for a phase I clinical trial, such a time-dependency may not be generalizable to the clinical settings in patients. Moreover, patients with dyslipidemia have a 30% higher baseline level of LDL-C than healthy subjects, which implies the difference in lipid metabolism between healthy subjects and patients [8].

## 5. Conclusions

In conclusion, the PK and PK/PD relationships of CKD-519 have been well characterized using the data from a multiple dosing study in healthy subjects. As a result of simulation from the final CKD-519 model, dosage regimens of 200–400 mg daily were recommended for further proof of concept studies in patients.

## Figures and Tables

**Figure 1 pharmaceutics-12-00573-f001:**
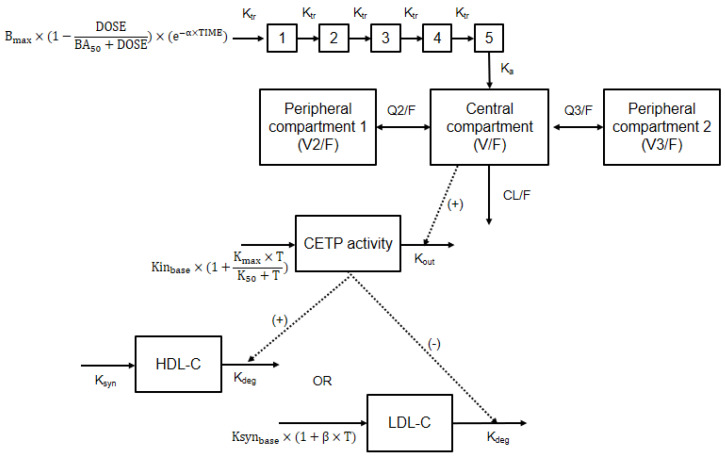
Schematic representation of the final pharmacokinetic and pharmacodynamics models for CETP inhibitor; The dose (DOSE) goes into the absorption compartment via transit compartments. It described by Erlang’s absorption model with transit rate constant (K_tr_). The fraction of drug that reaches absorption compartment is dose and time-dependent. The formation of CETP activity described by the first-order CETP activity elimination rate (K_out_), and also stimulated by the placebo effect via an E_max_ model. In addition, the elimination of CETP activity (K_out_) stimulated by plasma drug concentration (E_max_ model). The response of HDL-C or LDL-C described by the zero-order production and first-order elimination. Also, the CETP effect was incorporated into the model as a stimulatory function of HDL-C elimination and an inhibitory function of LDL-C elimination (sigmoid E_max_ model).

**Figure 2 pharmaceutics-12-00573-f002:**
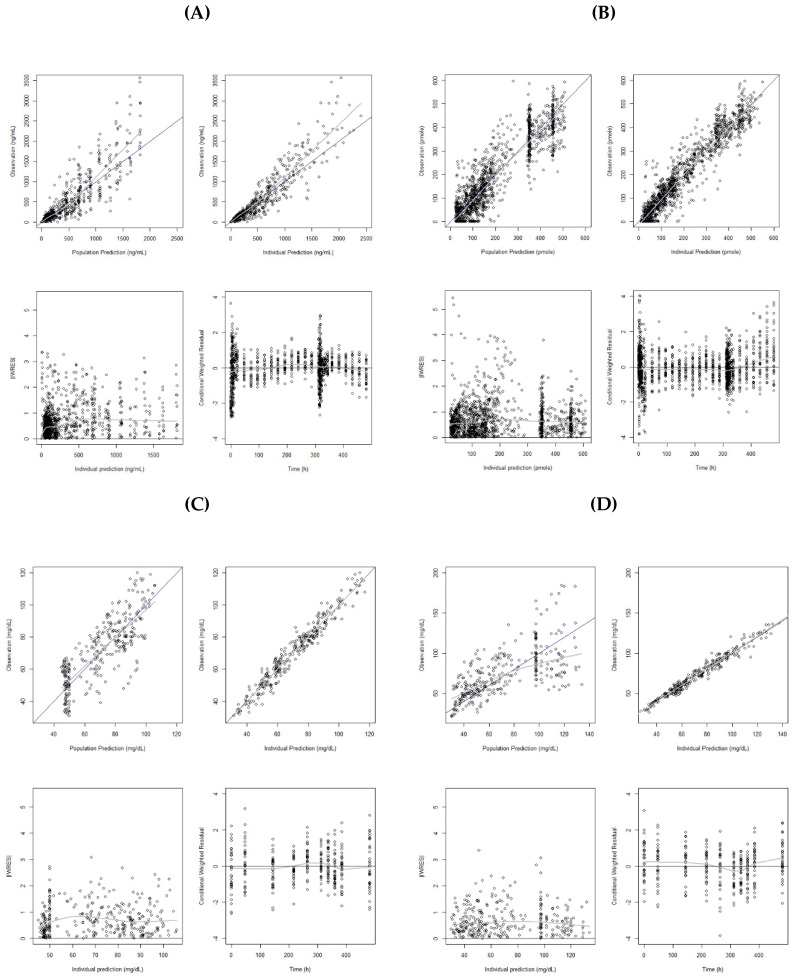
Goodness-of-fit plots for the final population pharmacokinetic and pharmacodynamic models: (**A**) Plasma concentration of CKD-519; (**B**) CETP activity; (**C**) HDL-C; (**D**) LDL-C.

**Figure 3 pharmaceutics-12-00573-f003:**
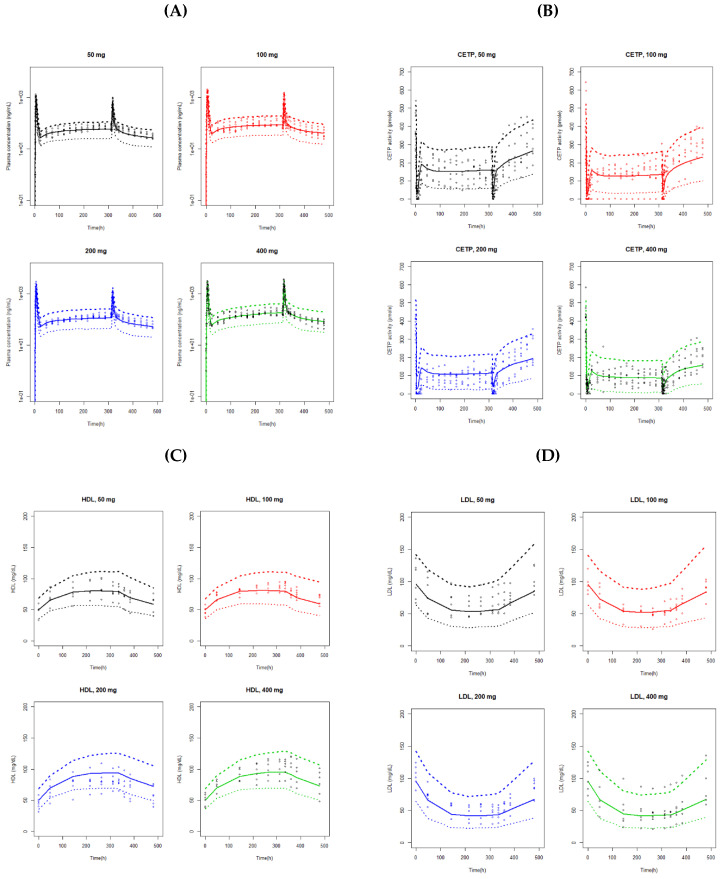
Visual predictive check plots for the final pharmacokinetic and pharmacodynamic models: (**A**) PK model; (**B**) CETP model; (**C**) HDL-C model; (**D**) LDL-C model. Open circles are the observations. The upper and lower dashed lines are the 95th and 5th percentiles of the simulated data based on 1000 simulations, respectively. The solid line is the median of the simulated data.

**Figure 4 pharmaceutics-12-00573-f004:**
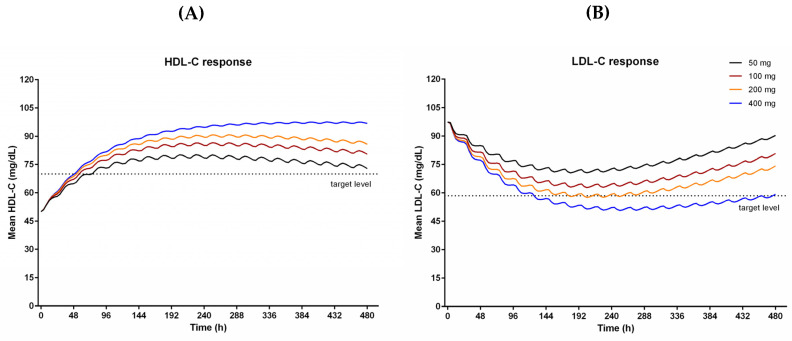
Simulation based assessment of the exposure-response relationship: (**A**) HDL-C response, (**B**) LDL-C response. They are simulated HDL-C and LDL-C responses after 21-day repeated administration of CKD-519. The dashed lines are the target values, which are set by a 40% change from baseline of HDL-C and LDL-C, respectively.

**Table 1 pharmaceutics-12-00573-t001:** Demographic characteristics of subjects at baseline.

Variables	50 mg	100 mg	200 mg	400 mg	Placebo	*p*-Value ^*^
	(*n* = 6)	(*n* = 6)	(*n* = 6)	(*n* = 6)	(*n* = 8)	
Age (years)	33.7 ± 5.5 (25.0–41.0)	34.0 ± 10.9 (21.0–47.0)	30.7 ± 9.9 (21.0–46.0)	30.3 ± 5.5 (22.0–39.0)	32.3 ± 9.7 (19.0–46.0)	0.924
Height (cm)	174.3 ± 7.8 (163.4–184.1)	174.4 ± 1.8 (172.1–176.6)	174.4 ± 6.7 (164.0–181.9)	175.1 ± 6.2 (167.7–184.7)	173.1 ± 4.9 (164.0–180.6)	0.980
Weight (kg)	69.7 ± 7.8 (59.9–76.8)	69.8 ± 1.6 (67.5–72.2)	71.5 ± 8.2 (58.3–79.5)	67.7 ± 7.1 (60.7–78.2)	65.9 ± 7.6 (58.5–79.5)	0.609
BMI (kg/m^2^)	22.8 ± 1.0 (21.8–24.3)	22.9 ± 0.7 (22.3–23.8)	23.4 ± 1.8 (20.3–24.9)	22.0 ± 1.9 (19.4–24.9)	21.9 ± 2.2 (19.0–24.3)	0.436
Cr (mg/dL)	0.79 ± 0.07 (0.69–0.86)	0.93 ± 0.05 (0.86–0.99)	0.89 ± 0.11 (0.75–1.02)	0.82 ± 0.09 (0.67–0.89)	0.84 ± 0.16 (0.67–1.14)	0.204
ALT (IU/L)	14.0 ± 4.7 (9.0–22.0)	19.3 ± 6.9 (14.0–33.0)	13.5 ± 5.1 (9.0–21.0)	13.3 ± 11.1 (5.0–35.0)	14.4 ± 7.3 (5.0–27.0)	0.600

Data are given mean ± SD (min–max). BMI, body mass index; CR, creatinine; ALT, alanine aminotransferase. **^*^**
*p*-value among dose groups were calculated with the ANOVA test.

**Table 2 pharmaceutics-12-00573-t002:** Parameter estimates from the final population pharmacokinetic model.

Parameter	Description	Estimate	%RSE	Bootstrap Median (95% CI) *
Structural Model				
CL/F (L/h)	Clearance	6.4	10.3	6.4 (5.2–8.1)
V/F (L)	Volume of central compartment	11.4	11.0	11.4 (9.4–15.1)
V2/F (L)	Volume of peripheral compartment 1	45.4	19.0	45.0 (23.2–72.2)
V3/F (L)	Volume of peripheral compartment 2	1006.0	32.4	1060.0 (458.0–2320.3)
Q2/F (L/h)	Inter-compartmental clearance for compartment 1	2.6	11.6	2.6 (1.9–3.6)
Q3/F (L/h)	Inter-compartmental clearance for compartment 2	3.3	15.4	3.3 (2.6–4.8)
K_a_ (h^−1^)	Absorption rate constant	1.09	3.5	1.09 (1.03–1.14)
K_tr_ (h^−1^)	Transit rate constant	1.10	3.4	1.10 (1.03–1.16)
F = BA × FT				
BA = B_max_ × (1 − DOSE/(BA_50_ + DOSE))				
B_max_	Maximal effect of dose on bioavailability	1.6	30.7	1.7 (0.7–11.4)
BA_50_ (mg)	Dose of half-maximal effect on bioavailability	90.1	24.5	87.8 (9.0–348.1)
FT = e^−α × TIME^				
α_1_	Proportionality constant for fraction of first-order absorption process depending on time for 50, 100, and 200 mg DOSE group	0.002	14.4	0.002 (0.001–0.002)
α_2_	Proportionality constant for thefraction of first-order absorption process depending on time for 400 mg DOSE group	0 FIX	NA	NA
Inter-Individual Variability (CV%)				
ω_CL/F_ (%)	Interindividual variability on CL/F	15.6	51.5	14.4 (0.3–29.2)
ω_V3/F_ (%)	Interindividual variability on V3/F	58.1	31.9	55.2 (20.0–77.2)
ω_Bmax_ (%)	Interindividual variability on B_MAX_	28.2	41.6	26.1 (14.6–38.4)
ω_Ktr_ (%)	Interindividual variability on Ktr	14.1	44.0	13.7 (6.4–19.5)
Residual Error				
σ_add_	Additive error	0.0001 FIX	NA	NA
σ_prop_	Proportional error	0.30	4.2	0.30 (0.27–0.32)

* 95% CI was estimated by applying the final population pharmacokinetic model to 1000 resampled datasets. %RSE, relative standard error; CI, confidence interval; F, relative bioavailability; CV, coefficient of variation; NA, not applicable.

**Table 3 pharmaceutics-12-00573-t003:** Parameter estimates for CETP model.

Parameter	Description	Estimate	%RSE	Bootstrap Median (95% CI) *
Structural Model				
CETP_base_ (pmole)	Baseline CETP activity	350.0	1.9	346.0 (315.0–372.0)
K_in_ = K_in,base_ × (1 + Placebo)				
Placebo = K_max_ × T/(K_50_ + T)				
Kin_base_ (pmole/h)	CETP activity increase rate constant at baseline	164.0	0.9	149.0 (33.0–345.1)
K_max_	Maximal effect of time on CETP activity production rate	9.6	2.2	9.8 (3.8–124.2)
K_50_ (h)	Time of half-maximal effect on CETP production rate	9700.0	17.3	8540.0 (3919.8–119650.0)
Drug = E_max_ × C_P_/(EC_50_ + C_P_)				
E_max_	Maximal effect of plasma drug concentration on CETP activity	18.2	0.8	20.0 (14.3–83.4)
EC_50_ (ng/mL)	Plasma drug concentration of half-maximal effect on CETP activity	587.0	3.7	597.0 (441.0–2581.0)
Inter-Individual Variability (CV%)				
ω_Kinbase_ (%)	Interindividual variability on Kin_base_	19.7	0.9	17.4 (9.9–26.3)
ω_Emax_ (%)	Interindividual variability on E_max_	39.5	0.5	41.1 (25.8–61.1)
Residual Error				
σ_add_	Additive error	40.4	4.2	38.0 (26.1–47.4)
σ_prop_	Proportional error	0.112	5.7	0.127 (0.058–0.237)

* 95% CI was estimated by applying the final population pharmacokinetic model to 1000 resampled datasets. %RSE, relative standard error; CI, confidence interval; Kin, production rate of CETP activity; Drug, drug effect on CETP activity; C_P_, plasma drug concentration; CV, coefficient of variation; NA, not applicable.

**Table 4 pharmaceutics-12-00573-t004:** Parameter estimates for HDL-C and LDL-C model.

Parameter	Description	Estimate	%RSE	Bootstrap Median (95% CI) *
HDL-C				
Structural Model				
RB (mg/dL)	Baseline HDL-C	50.0	13.2	50.0 (46.3–53.5)
Ksyn (h^−1^)	Production rate of HDL-C	1.26	15.2	1.26 (1.03–1.57)
RES = (CETP^γ^ × R_max_)/(CETP^γ^ + R_50_^γ^)				
γ	Steepness of the CETP activity versus HDL-C relationship	2.1	5.3	2.1 (1.1–3.6)
R_max_	Maximal effect of CETP activity on HDL-C response	1.5	56.5	1.6 (1.1–3.4)
R_50_ (pmole)	CETP activity of half-maximal effect on HDL-C response	185.0	24.8	185.0 (143.0–390.1)
Inter-individual variability (CV%)				
ω_RB_ (%)	Interindividual variability on RB	17.2	29.3	16.8 (12.2–21.3)
Residual Error				
σ_add_	Additive error	4.6	6.2	4.5 (4.0–5.0)
σ_prop_	Proportional error	0.0.0001 FIX	NA	NA
LDL-C				
Structural Model				
RB (mg/dL)	Baseline LDL-C	97.4	1.9	94.3 (83.4–105.0)
Ksy = Ksyn × (1 + β × T)				
Ksyn (h^−1^)	Production rate of LDL-C	0.435	1.5	0.419 (0.245–0.573)
β	Proportionality constant for fraction of LDL-C production rate	0.0009	0.8	0.0008 (0.0004–0.0021)
RES = (CETP^γ^ × R_max_)/(CETP^γ^ + R_50_^γ^)				
γ	Steepness of the CETP activity versus LDL-C relationship	2.2	1.3	1.9 (0.9–3.4)
R_max_	Maximal effect of CETP activity on LDL-C response	0.8	1.1	0.8 (0.8–1.0)
R_50_ (pmole)	CETP activity of half-maximal effect on LDL-C response	80.7	3.5	68.9 (38.8–107.0)
Inter-Individual Variability (CV%)				
ω_RB_ (%)	Interindividual variability on RB	23.6	36.8	23.7 (18.0–31.3)
ω_β_ (%)	Interindividual variability on β	97.4	14.1	102.8 (53.6–239.8)
ω_GAM_ (%)	Interindividual variability on γ	50.9	1.5	48.4 (20.8–90.6)
Residual Error				
σ_add_	Additive error	0 FIX	NA	NA
σ_prop_	Proportional error	0.07	1.8	0.07 (0.06–0.08)

* 95% CI was estimated by applying the final population HDL-C or LDL-C model to 1000 resampled datasets; %RSE, relative standard error; CI, confidence interval; RES, HDL-C or LDL-C response by CETP activity; CETP, CETP activity; CV, coefficient of variation; NA, not applicable.

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
