# Peer review of "A Population Pharmacokinetic and Pharmacodynamic Model of CKD-519"

_pharmaceutics, 2020, doi:10.3390/pharmaceutics12060573_

Round 1
Reviewer 1 Report
Dear Editor,
Thank you very much and it is an honor to serve the role.
Personally, I think this article is well written and presented.
However, since this is a Population PK/PD modeling exercise, it is essential to have author to disclose information such as compound structure, physicochemical data, and other in vitro data (i.e Papp) to the public. The above information is particular important since author mentioned “It exhibited a maximum of 63–83% inhibitory effect of CETP activity after single administration with 25–400 mg in healthy subjects (EC50 17.3 ng/mL). The exposure of CKD-519 increased with the dose. However, maximum plasma concentration (Cmax) and area under the plasma concentration-time curve (AUC) normalized by dose decreased with each incremental dose [4]. In addition, the pharmacokinetics (PK) of CKD-519 exhibited significant variation in the fasted and fed states. Administration of single CKD-519 200 mg with standard meal resulted in 8.7 and 5.8 fold 47 increase in Cmax and AUC relative to the fasted state, respectively (in house data). The inhibitory effect of CETP activity by CKD-519 200 mg was also higher in the fed (91.4%) than in the fasted group (81.5%)”.
Without more information, reader cannot fully picture the exercise and the accuracy of the model.
Thanks again
Tom
Author Response
Thank you for your valuable comments. As your opinion, we added CKD-519 physicochemical data in the manuscript (Page 1; Line 41–44).

Reviewer 2 Report
Manuscript entitled "A Population Pharmacokinetic and Pharmacodynamic
Model of CKD-519" by Authors Choon OK Kim et al is done extensively with 4 dose levels for PK/PD correlation. Results are presented in detail.
minor comments:
- Introduction needs to be little more about pharmacology of CKD-519 including pharmacokinetics in fed and fasted state, various dose levels (already available data)
- Methodology: How much blood was collected per time points, since many time points were collected for PK analysis
- What was the anti-coagulent used for plasma separation.
- Please provide brief method about LCMS/MS analysis of CKD-519
- Results: Figure 2 is missing.
- PMID - 27895466 entitled "Pharmacokinetics, pharmacodynamics and safety of CKD-519, a CETP inhibitor in healthy subjects" also indicated 400mg dose is safe and potent. How was this results are different from that old results.
Overall the study is extensive.
Author Response
Thank you for your thoughtful review comments.
We replied on your comments that needed to be answered or to be revised. We attached response file as below.
